# The Impact of Hormesis, Neuronal Stress Response, and Reproduction, upon Clinical Aging: A Narrative Review

**DOI:** 10.3390/jcm12165433

**Published:** 2023-08-21

**Authors:** Marios Kyriazis, Lama Swas, Tetiana Orlova

**Affiliations:** National Gerontology Centre, Larnaca 6043, Cyprus

**Keywords:** hormesis, neuronal stress response, neuron-germline communication, neuron-germline conflict, cognitive challenges, enriched environment, age-related degeneration

## Abstract

Introduction: The primary objective of researchers in the biology of aging is to gain a comprehensive understanding of the aging process while developing practical solutions that can enhance the quality of life for older individuals. This involves a continuous effort to bridge the gap between fundamental biological research and its real-world applications. Purpose: In this narrative review, we attempt to link research findings concerning the hormetic relationship between neurons and germ cells, and translate these findings into clinically relevant concepts. Methods: We conducted a literature search using PubMed, Embase, PLOS, Digital Commons Network, Google Scholar and Cochrane Library from 2000 to 2023, analyzing studies dealing with the relationship between hormetic, cognitive, and reproductive aspects of human aging. Results: The process of hormesis serves as a bridge between the biology of neuron-germ cell interactions on one hand, and the clinical relevance of these interactions on the other. Details concerning these processes are discussed here, emphasizing new research which strengthens the overall concept. Conclusions: This review presents a scientifically and clinically relevant argument, claiming that maintaining a cognitively active lifestyle may decrease age-related degeneration, and improve overall health in aging. This is a totally novel approach which reflects current developments in several relevant aspects of our biology, technology, and society.

## 1. Introduction and Methods

From a clinical viewpoint, aging is a process which progressively diminishes the function of the person, resulting in chronic degenerative conditions, until this burden becomes incompatible with life. In this respect, one of us (Kyriazis, 2020) has defined aging as: ‘Time-Related Dysfunction’ [1], with the explanation that:


*“This definition implies that, with the passage of time and for a variety of causative factors, humans are subjected to damage which is not properly repaired. As a consequence, there is degeneration and loss of utility at all levels (molecular, cellular, tissue, organismic, and societal) with a resulting failure of the normal function of a human. In other words, it is a chronologically-dependent erosion of our functions, which makes it increasingly difficult for us to manage and operate within a given, always-changing environment”.*


Therefore, encouraging specific clinical interventions may positively influence time-related dysfunction and diminish the rate of decline. One such intervention is the use of digital technology which, through the neuronal processing of meaningful information, up-regulates the function of our brain, and may prevent age-related degeneration. Details of this process are discussed below.

Our everyday life (at least in developed countries) is progressively becoming less physical and more cognitive [2]. We are exposed to an environment that is becoming less of a combination of several interactive organisms but, increasingly, more of an environment of only two elements: humans and digital machines [3]. This interaction exposes us to an increased cognitive burden, which may affect the way our brain operates. It may redefine the existing energetic balance that exists between neuronal cells and the germline. In this paper, the term ‘neuronal cells’ or ‘neurons’ refers to glial cells, oligodendrocytes, astrocytes, intermediate progenitors, immature neurons, mature neurons, glutamatergic neurons, GABAergic neurons, dopaminergic neurons, cholinergic neurons, and motor neurons. The term ‘germline’ is explained as: “cells, such as spermatozoa, ova or spermatogonial stem cells (or primordial germ cells), which participate in the process of transmission of genetic material to the progeny” [4].

There is a process of both direct and indirect communication between somatic cells (specifically, neurons), and germline cells [5,6,7,8,9]. The immense relevance of the presence of these conserved pathways of communication will become clear further on in this paper. The increasing cognitive/digital nature of our environment is slowly altering the survival relationship between neurons and germline [10]. This relationship is grounded in hormetic notions which explore the effects of ‘stimulation’ (a challenge that disturbs homeostasis) and the biological response to this stimulation. The concept of hormesis and its relationship with neuronal and germline survival is relevant here, and will be discussed below.

In this narrative review we examined publications listed in PubMed, Embase, PLOS, Digital Commons Network, Google Scholar and Cochrane Library from 2000 to 2023 for the terms ‘hormesis’, ‘hormesis in aging’, ‘hormesis in health’, ‘germline survival’, ‘neuronal stress response’, and ‘neuron/germline communication’. Our aim is to examine relevant (albeit sometimes seemingly unrelated) concepts, in one narrative paper describing the complex relationship between cognition, reproduction, and aging.

We used a generative pre-trained transformer (ChatGPT-4) to generate a small amount of original background text, in less than 5% of the paper. The text suggested by GPT-4 has been verified and methodically checked. In addition, GPT-4 has been used in places (Section 2.1, Section 2.4 and Section 2.5), in order to support ideation and concept checking of the manuscript. The feedback originated by ChatGPT-4 was discussed, modified, and verified by two of us (M.K. and T.O.).

## 2. Results and Discussion

### 2.1. Hormesis

During the past several years, there have been many attempts to understand the process of hormesis [11]. Hormesis is a phenomenon in which a low dose of a stressor, such as a chemical or radiation, can stimulate a beneficial response in an organism. Several researchers have proposed that hormesis plays a role in the process of aging [12,13,14]. Hormesis is present in many aspects of the biological world where there is ‘activation’ of a function during application of a low-dose stimulus, and ‘inhibition’ if the dose is increased beyond a certain threshold. It is a non-linear, ‘inverted U’-shaped relationship between dose–effect (Figure 1). Therefore, the principle underlying hormesis is relatively simple: Low dose of a stimulus can positively challenge the organism eliciting a controlled stress response, and result in health benefits, whereas an excessive, suboptimal, or prolonged exposure to the same stimulus can result in damage and disease [15,16]. It should be emphasized that several researchers consider hormesis as a controversial subject, because the original research on hormesis and radiation has not been proven. Nevertheless, an increasing number of other experts, study hormesis as a relevant and valuable subject in health and, particularly, in aging.

Single or multiple exposure to low doses of otherwise harmful agents, such as irradiation, food restriction, heat stress, hypergravity, reactive oxygen species, and other challenges, has a variety of anti-aging and longevity-extending effects [17]. Detailed molecular mechanisms that bring about the hormetic effects are still not very clear but are being increasingly understood, and comprise a cascade of stress response and other pathways of maintenance and repair [18].

Although the extent of immediate hormetic effects after exposure to a particular stress may only be moderate, the chain of events following initial hormesis leads to biologically amplified effects that are much larger, synergistic and pleiotropic. A consequence of hormetic amplification is an increase in the homeodynamic space (Box 1) of a living system in terms of increased defense capacity and reduced load of damaged macromolecules [19]. Hormetic strengthening of the homeodynamic space provides wider margins for metabolic fluctuation, stress tolerance, adaptation and survival. Hormesis thus counter-balances the progressive shrinkage of the homeodynamic space, which is the ultimate cause of aging, diseases and death. Healthy aging may be achieved by hormesis through mild and periodic (but not severe or chronic) physical and mental challenges [16], and by the use of nutritional hormesis incorporating mild stress-inducing molecules called hormetins. The increasingly established scientific foundations of hormesis are ready to pave the way for new and effective approaches in aging research and intervention.

Thus, hormetic stimuli can be nutritional (dietary restriction, intermittent fasting), physical (exercise, heat, cold), chemical (nutritional or pharmaceutical compounds, hormetins), and mental (brain exercises, meditation, cognitive ‘positive-stress’). These may cause a slight injury to the organism, i.e., they disturb homeostasis, thus activating stress response pathways which aim to restore homeostasis, and up-regulate repair mechanisms. During the process of repairing this hormetically-induced damage, any coincidental age-related damage may also be repaired [20].

Box 1What is the homeodynamic space?
Rattan explains the concept of ‘homeodynamic space’ as encompassing “three characteristics: stress response, damage control, and constant remodeling, which provide measurable biomarkers reflecting the survival ability, robustness, and resilience of a biological system. A biological definition of health thus involves measures of functionality, tolerance and adaptation” [21].The homeodynamic space refers to the range of physiological states and processes that an organism can tolerate and maintain without suffering harm or death. This range is determined by the organism’s capacity to adapt to and cope with stress [22].The concept of homeodynamic space is related to the concept of homeostasis. However, while homeostasis refers to a narrow range of physiological states that are optimal for survival, homeodynamic space refers to a much broader range of physiological states and processes that an organism can tolerate. It highlights the dynamic nature of physiological systems and the importance of resilience and adaptability in maintaining health and well-being.


Hormesis may, for example, be involved in the process of calorie restriction, which has been shown to extend lifespan in animals [23,24]. Calorie restriction has been proposed to work, in part, by causing mild stress in cells, which may stimulate the production of stress-response proteins and other molecules that have anti-aging effects [25].

The role of hormesis within the context of aging is thought to be very relevant, particularly when we consider neuronal hormetic stimulation. Exposure to new information, most likely derived from digital sources (internet, social media) maintains our neurons in a state of novelty [26], creating positive effects on health. This activation of neuronal stress response follows hormetic exposure (to meaningful ‘information-that-requires-action’) and thus up-regulates neuronal health [27].

Despite the promising evidence for the role of hormesis in aging, it is important to note that the mechanisms by which it may work are not fully understood, and more research is needed to determine the extent to which hormesis is involved in the aging process. It is also important to note that hormesis is a complex phenomenon, and the effects of any given stressor will depend on the specific dose, duration, and timing of exposure, as well as the genetic and environmental background of the organism.

A crude way to establish if clinical hormesis is taking place:


*A basic characteristic of a hormetic event is the novelty of information. Hormesis is present when the challenge is of sufficient magnitude and appropriate quality as to satisfy the definition of ‘novelty’. Novelty is defined as ‘the quality of being new, original, or unusual’, and this includes both unfamiliarity and unconventionality. Routine and monotony generally do not invoke a hormetic response. The assessment and response to the new challenge leads to adaptation and thus, eventually, improvement of function within a particular environment (the environment where the challenges have originated from). Therefore, when a certain stimulus appears ‘novel’ to us, then it is likely that this stimulus is eliciting a hormetic response. When the stimulus appears boring or monotonous, it is less likely that hormesis is taking place.*


### 2.2. Autophagy and Hormesis

Autophagy is one of the markers of hormesis, i.e., the presence of efficient autophagy indicates that hormesis is occurring, and enhancing the processes of biological repair. Autophagy is a protein turnover pathway, a catalytic process, which aims to degrade and recycle cellular components. This process maintains cellular function during (or after) stress, when damaged material accumulates and it has to be eliminated [28].

The process of autophagy can be enhanced via hormetic stress, such as:Exercise. This can enhance autophagy in liver, muscles, pancreas and adipose tissue, as well as in the brain [29].Moderate hot/cold exposure [30] via activation of Heat Shock Proteins (HSP) [31]. Hormetic stress in addition to HSP involvement, also reduces the progressive accumulation of PolyQ aggregates [32].Intermittent fasting (IF) is a nutritional hormetic stress. Alirezaei et al. [33] conducted a study to investigate the effects of food restriction and short-term fasting on autophagy. Their findings revealed that food restriction induces autophagy in mouse livers, challenging the conventional belief of the brain’s metabolic privilege. Moreover, their research suggests that sporadic fasting could be a cost-effective approach to promote a therapeutic neuronal response. In a separate study, Pietrocola et al. [34] emphasized the significance of autophagy in cancer treatment. They highlighted that impairment of autophagy reduces the effectiveness of chemotherapy and radiotherapy. These findings underscore the importance of understanding autophagic mechanisms to enhance cancer treatment strategies. Additionally, Kim and Lemasters [35] observed the occurrence of autophagy in liver cells during fasting, providing further insights into its role in cellular recycling. Their study demonstrated that liver cells form phagophores and autophagosomes, which encapsulate and capture mitochondria for recycling. This process leads to the breakdown of mitochondria and their contents, including DNA. In addition to physical stimuli, autophagy can be modulated by hormetins, i.e., substances that can induce health-beneficial physiological hormesis [36] and this is an appropriate opportunity to discuss some more details of hormetins (Table 1).

There are specific stress-induced pathways for enhancing autophagy in neurons [52,53], and it is known that the stress response activates autophagy [54,55,56]. This shows the direct relationship that exists between autophagy and stress. It is therefore reasonable to infer that, if this wide range of hormetic stresses improves autophagy, it could well be that other hormetic stresses, such as a cognitive stress may also have similar effects.

### 2.3. Environmental Enrichment

A concept relevant to hormesis is environmental enrichment (EE). This refers to a varied and stimulating environment that promotes physical and psychological well-being. It involves creating an environment that is stimulating and challenging, both physically and mentally. This can include activities such as exercise, social interaction, and learning new skills, as well as exposure to novel and varied stimuli, such as music, art, and digital information that requires a response [26]. The concept of EE has gained prominence in recent years as increasingly more research has demonstrated its benefits for both animals and humans. It can improve cognitive function, reduce stress, and promote natural behaviors. It also improves mood, reduces stress and anxiety, and increases overall quality of life [57].

Studies have shown that animals raised in enriched environments demonstrate better learning and memory, as well as improved problem-solving abilities [58,59]. This is likely because the complexity of an enriched environment provides greater opportunities for cognitive stimulation and growth. In the case of aging, environmental enrichment is becoming increasingly recognized as an effective tool for promoting healthy aging and, specifically, improving cognition [60].

Cognitive stimulation is another key component of environmental enrichment for older adults. This can include activities such as learning a new language, playing a musical instrument, engaging in brain-training exercises, or other virtual or digitally-derived cognitive activities. These activities can help keep the brain active and engaged, promoting cognitive function and reducing the risk of cognitive decline [61]. The new information reaching the brain acts as a hormetic stimulus or a challenge, that activates the neuronal stress response and requires the brain to act in order to deal with this new challenge, through remodeling and increase robustness [26].

It was shown that an environment which is rich in cognitive stimuli, has indirect effects on tissues and organs other than the brain. For instance, some authors have argued that an enriched environment improves vision [62,63], while others reported the benefits of a cognitively enriched environment on:Immunity [64];Wound healing [65,66];The retina [67];Muscle strength, without the need to physically exercise (!) [68];Inflammatory response [69,70] and other physical parameters [71], such as vitality, physical functioning and bodily pain, as well as social and emotional functioning [72]. Many of these effects may persist for several years, in some cases even after a 10 year period [73].

### 2.4. Neuronal Stress Response

The neuronal stress response is the set of molecular and cellular changes that occur in neurons in response to stress or injury [74]. These changes help neurons adapt to, and survive, stress, and they can also have important consequences for the function and health of the nervous system [75]. One of the key factors in the neuronal stress response are the stress-response proteins. These proteins are activated in response to various stressors, including heat, cold, radiation, and certain chemicals, as mentioned above. Once activated, stress-response proteins support neurons overcome stress by modifying their gene expression, protein synthesis, and other cellular processes [76,77], such as ATP generation in times of stress [78] and the modulation of the signaling molecule cyclic AMP [79]. It is important to note that the neuronal stress response is not a uniform process, and different neurons may respond to stress in different ways depending on their specific function and location in the nervous system. Additionally, the response to stress can vary depending on the severity and duration of the stressor, as well as the genetic and environmental background of the organism.

### 2.5. Digital Information, Cognition, and Neuronal Stress Response

The advent of information technology has brought about a significant rise in the cognitive load imposed on our brains, primarily due to the sheer volume of information we now encounter [80]. The internet and social media platforms offer us access to an overwhelming abundance of information, making it increasingly difficult to sift through and identify what is truly important and relevant. As a result, individuals often struggle to concentrate on productive and meaningful tasks as they grapple with the challenge of filtering out the noise and distractions surrounding them [81].

This can lead to cognitive overload, which can impair cognitive function and lead to feelings of stress and fatigue. Our neurons are subjected to the phenomenon of ‘neuronal fatigue’ and they stop responding to an unchanged, continual monotonic stimulation. Such a stimulation causes the neuron to lose its ability to transmit activation to other neurons [82]. On the other hand, a moderate (in other words, hormetic) amount of information that requires us to act, may impact positively on the brain, up-regulating the neuronal stress response and thus enhancing the robustness of neuronal function [83]. In essence, we are living in an enriched environment, as described above.

We know that digital cognitive training improves cognition and may reduce the risk of dementia [84]. Studies have repeatedly shown that ‘serious games’ have a positive impact on dementia patients [85,86]. ‘Serious games’ are participative digital/electronic games designed for purposes other than entertainment. Specifically, Yang et al. [87] state that there is:


*“…evidence that video game interventions could be considered for the elderly for improving performance and cognitive function, especially general cognitive scores and processing speed. Games with better interactivity and visual stimulation have better curative effects…”.*


In addition, electronic games used generally for entertainment also have positive effects on the memory of older people [88,89].

By being exposed to a judicious, ever-changing, novel and positive amount of information, it becomes necessary for our neurons to acquire additional repair resources and thus function for longer, with a consequent overall improvement in healthy lifespan. These additional energetic resources are subjected to a trade-off: as a balancing (trade-off) measure, germline repair mechanisms need to be down-regulated to accommodate a corresponding escalation of repairs in neurons [90]. This is because there is a close and very relevant relationship between neurons and germline cells that will be explored in detail below.

### 2.6. The Bidirectional Communication (Cross-Talk) between Neurons and the Germline

Some years ago, Ermolaeva et al. [91], suggested that genetic injury in germline cells, may act as a stimulus to initiate protective effects in somatic cells. In other words, elements in the germline up-regulate the function of somatic cells. This may happen through up-regulation of the stress resistant mechanisms in such somatic cells. Others have confirmed and elaborated on this [92,93]. Khodakarami et al. [94] have suggested that this germ-initiated somatic protective mechanism reveals a conserved tendency to reverse the trade-offs that exist between germ cell and somatic cell repair.

Data increasingly suggest that there is open communication between the soma (i.e., all cells in the body which are not involved in reproduction—here, specifically the neuron) and the germline [95,96,97]. Information is transferred through ‘cross-talk’ from the soma to the germline (and in reverse), and this information may negatively affect the aging of the germline or the aging of the soma [95,98]. There is an increasing indication to show that the process of resources flowing from the soma to the germline is not unidirectional. It is possible to experience the reverse, whereby resources could move from the germline back to somatic cells, up-regulating their function [99,100]. We also know that there exist carriers of epigenetic information from the soma to the germline including microRNAs or even extracellular vesicles which move from the soma to the germline environment [101,102].

Some examples of neuron-to-germline communication are described below, with a summary in Box 2.

DNA damage in germ cells increases resilience in somatic cells via the ERK MAP kinase MPK-1 pathway [91]. Furthermore, when somatic cells experience stress, there is an increased demand (by somatic cells) for repair resources, which are diverted from the germline [90].Germline cells have the innate capacity to become neurons following suitable natural (or artificial) reprogramming by transcription factors, even though there are several conserved mechanisms that safeguard against this. This is an intriguing situation because it shows the direct relationship between the germline and the brain [103]. In addition, multipotent neural and glial precursors can be derived from multipotent adult germ line stem cells [104]. These multipotent neural precursors are able to mature and integrate within the existing neural network. It is necessary to mention that, although these effects have been found in experiments conducted in vitro, clinical tests in vivo are still lacking. It is, however, remarkable to realize that the germline acts as a source of fully functional neurons [105,106,107].More specifically, we know that germline (spermatogonial) stem cells may act as a source of neuron-like cells [108], and definitive neural stem cells [109].A more detailed direct communication pathway between germline cells and the soma has been studied by Levi-Feber et al. [110] who showed that this pathway depends on the endoplasmic reticulum stress factor inositol requiring enzyme-1 (IRE-1).On certain situations, ectopic germline cells can be found in the brain, and could contribute to altered neuronal development, resulting in neurodevelopmental disorders. This indicates not only the close relationship between neurons and germ cells, but also the continual struggle for equilibrium, between these two [111].Furthermore, there is another fact that underlines the close relationship between neurons and germ cells. Progesterone, which modulates sperm function, acts (via intermediate steps) by interacting with “a sperm membrane receptor which resembles the neuronal GABA(A) receptor” in the brain [112].It was shown that eradication of germ cells in Drosophila, has a positive impact on its lifespan, possibly through modulation of the nutrient sensing insulin/insulin-like (IIS) growth factor signaling [113]. This strengthens the general argument that that somatic lifespan is under the control of the germ line, and vice versa.The repressor element 1-silencing transcription factor REST which modulates multipotent stem cells, is present in testes, but, intriguingly, regulates target genes in neurons [114]. REST activity has been associated with cognitive impairment and dementia, whereas a potent activity of REST is associated with modulating the balance of neuroprotection vs. neurodegeneration—i.e., acts in a hormetic way [115,116]. Thus, there exist a conserved mechanism of modulation of neural development regulated by REST which is present in spermatogonial cells, indicating another possible mechanism of neuro-germline communication.It is known that the germline may influence the function of distant somatic cells, including neurons. For instance, germline stem cells influence proteostasis and thus prevent abnormal protein accumulation in neurons [117]. Thus, at least theoretically, the risk of neurodegenerative diseases is reduced.Spermidine is a natural polyamine compound with effects on heart disease, brain degeneration, cancer, and inflammation, among others. It also extends lifespan and health span [43], and it modulates autophagy both in the germline and in the neuron [118]. It was originally found in sperm and this begs the question: how and why does a compound in semen benefit the neuron? Spermidine is a mediator of the complex relationship that exists between neurons and the germline. The concentration of spermidine (apart from its high concentration in the sperm) is also high in the human brain. This must be because it has important actions to perform there [119]. It has positive actions on neuronal mitochondria [118], improves autophagy in neurons [120], and in germline stem cells [121], protects against synaptic degeneration [122] and exhibits general neuroprotective actions [123]. There are studies linking consumption of spermidine with a reduced risk of cognitive impairment in humans [124].

Box 2Summary of some examples of neuron-to-germline communication (References are given in the text description).
DNA damage in germ cells increases resilience in somatic cells [91].Neuronal stress causes an increased demand (by neurons) for repair resources, which are diverted from the germline [90].Germline cells have the capacity to become neurons. Neural precursors from the germline are able to mature and integrate within the existing neural network [103,104,105,106,107].A direct communication pathway between germline cells and the soma depends on the endoplasmic reticulum stress factor inositol requiring enzyme-1 (IRE-1) [110].Ectopic germline cells can be found in the brain, and could contribute to altered neuronal development, resulting in neurodevelopmental disorders [111].Progesterone, which modulates sperm function, acts by interacting with a membrane receptor which resembles the neuronal GABA(A) receptor in the brain [112].Eradication of germ cells in Drosophila, has a positive impact on its lifespan, possibly through modulation of the nutrient sensing insulin/insulin-like (IIS) growth factor signaling [113].The repressor element 1-silencing transcription factor (REST) which modulates multipotent stem cells, is present in both neonatal and adult testes, and regulates target genes in neurons [114].There is a conserved mechanism of modulation of neural development regulated by REST which is present in spermatogonial cells [115,116].The germline may influence the function of distant somatic cells, including neurons. For instance, germline stem cells influence proteostasis and thus control abnormal protein accumulation in neurons [117].We mention the example of the hormetin spermidine, which modulates autophagy both in the germline and in the neuron [43,118,119,120,121,122,123,124].


### 2.7. Neurons vs. Germline

There is a balance between allocation of resources to the different organs in any given organism. These resources could be allocated for damage repair or for growth of the organism [125]. Specifically, at the current stage of human evolution, nature has a propensity to favor allocation of repair resources to the germline [126], in order to assure the survival of the species, even if this means that allocation of resources to other organs (including the brain) will have to be suboptimal [127,128].

Some years ago, we proposed the Indispensable Soma Hypothesis (www.indispensablesoma.info accessed on 28 June 2023). We suggested that there is a direct competition for survival between neurons (cognition) and the germline (reproduction) [129], where neurons try to survive and function well by diverting repair resources from the germline. This means that a healthy neuron may live longer (and therefore we too live longer), when at the same time, the germ cells remain without adequate repairs, become defective and this results in a reduced reproduction. It may be possible to manipulate this relationship through hormetically increasing the function of the neurons, and thus be able to reduce age-related degeneration [130].

The suggestion that energetic trade-offs exist between organs that are costly to repair and others that are less costly, has been made several years ago, under the term ‘The Expensive Tissue Hypothesis’ [131]. Owing to the fact that repair resources are finite, there are preferred energy investments in organs that are evolutionarily ‘important’ followed by reduced investments in other organs [132]. Several studies have supported this general principle [133,134,135].

Here, the general meaning of the term ‘trade-off’ is taken to be: improvement in one aspect tends to be counterbalanced by deterioration in another aspect.

As an extension of this general hypothesis, a more specific hypothesis has been suggested: ‘The Expensive Germline Hypothesis’ [136]. Evidence supports the view that germline maintenance is costly (the expensive germline hypothesis) and that there are direct trade-offs with somatic maintenance [90]. This is why time-related degenerative damage to all organs is not repaired properly (and we have loss of function, age-related degeneration and death), whereas damage to the germline is as optimal as it could be [137].

However, we are now witnessing a general shift from this situation. Due to the fact that there is so much useful, relevant and actionable information reaching our brains (via sharing of digital information), we are witnessing, for the first time in human history, a shifting of priorities: from the germline, to the neuron (from the survival of the species, to the survival of the individual) [138]. By redirecting resources from the germline, neurons are able to maintain their structural and functional integrity over a long period of time. This process is thought to involve the up-regulation of certain genes and pathways that are involved in the maintenance and repair of neurons, as well as the down-regulation of genes and pathways that are involved in the production of gametes.

It is important to note that this process of resource redirection is not unique to neurons, and many other types of cells are also able to redirect resources from the germline in order to maintain their structural and functional integrity. However, the specialized nature of neurons and their long lifespan make them particularly dependent on this process.

## 3. Conclusions

In this paper, we essentially make the first steps in describing a new biology. A biology not based on reproduction and aging, but based on cognition and indefinite survival without chronic degenerative diseases. It represents a shift from a physical model of aging to a more cognitive one.

Specifically, it appears increasingly relevant that there is a connection between brain function and longevity. This is a complex process, influenced by a large number of factors. Germline elements may transmit somatic factors to somatic cells in order to increase somatic function. The success of this process depends on hormetic constraints (“too little is bad, too much is also bad”).

Our current enriched environment depends less on physical abilities, and more on cognitive ones. The continual exposure to new, meaningful, digitally-derived information that requires us to act, has typical hormetic characteristics, whereby neurons are challenged by this input of information, and need to adapt in order to process it. This process of adaptation takes place at the expense of resources allocated to the germline, which are diverted to the neuron, in order to execute successfully the neuronal stress response. The balance of repair resources shifts from the germline to the neuron, resulting in reduced resources for reproduction but increased somatic (neuronal) repairs, leading to reductions in age-related degeneration, longer, healthier lifespans and reduced reproduction rates, just as we increasingly see in developed, technological societies. A schematic representation of these general concepts is given in Figure 2.

Based on the discussion above, there is one important piece of clinical advice we can give to the public: We should intentionally expose ourselves to meaningful and novel digitally-derived information, information that requires us to act constructively and creatively, in other words, maintain our brain in a state of hormetic ‘positive stress’. Hormesis is an important concept here. There is solid recent research describing the role of hormesis in health [139,140,141,142,143,144,145,146] and also studies concentrating specifically on hormesis in aging—in addition to those studies already mentioned above [147,148,149,150,151,152,153].

An effort should be made to use digital technology as a means of enhancing our cognitive abilities, which will eventually reflect on a reduction in age-related degenerative diseases, through diversion of repair resources from the germline to somatic cells, and particularly, to the neuron.

## Figures and Tables

**Figure 1 jcm-12-05433-f001:**
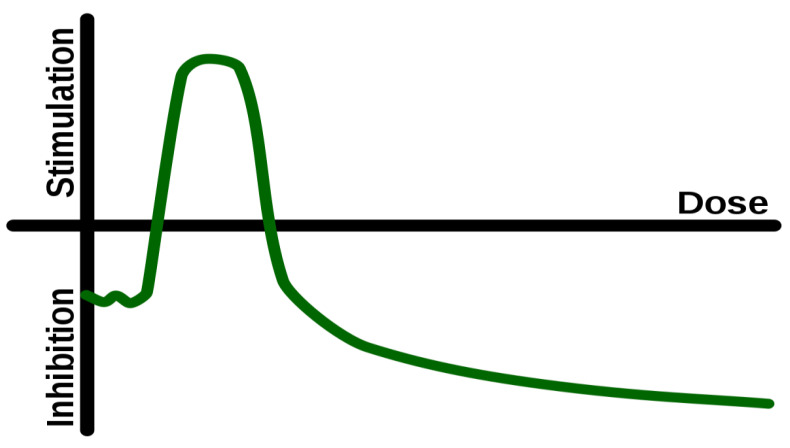
Hormesis: After an initial inhibition at a very low dose, as the dose increases there is a ‘window’ of stimulation, followed again by inhibition at higher doses.

**Figure 2 jcm-12-05433-f002:**
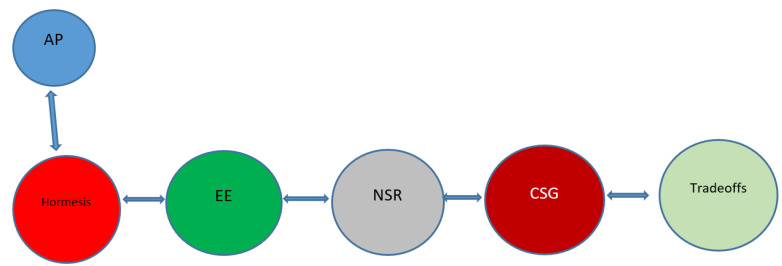
Graphical summary of the subjects discussed. The logical progression of the discussion of these, perhaps seemingly unrelated subjects is schematically depicted below. The discussion starts from the concept of hormesis (and the example of autophagy (AP), and then progresses to the subjects of environmental enrichment (EE), neuronal stress response (NSR), the communication between somatic and germline cells (CSG), and the war of trade-offs between the neuron and the germline. The result may be an improvement of the clinical parameters of the aging patient.

**Table 1 jcm-12-05433-t001:** Examples of hormetins and their main actions.

Hormetin Supplements	Description	Reference
Rhodiola	Adaptogen, antioxidant	[37,38]
Schisandra	Adaptogen, liver conditions, tonic	[39,40,41,42]
Spermidine	Biological modulator, longevity (See section on spermidine)	[43,44,45]
Caffeine	Cognitive enhancer	[46]
Ginger	General health	[47,48]
Turmeric (Curcumin)	Anti-inflammation, antioxidant	[49,50,51]

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
