# Peer review of "The Impact of Hormesis, Neuronal Stress Response, and Reproduction, upon Clinical Aging: A Narrative Review"

_jcm, 2023, doi:10.3390/jcm12165433_

Round 1
Reviewer 1 Report
First of all, I want to express my gratitude to the authors for the opportunity to get acquainted with such an interesting article. The article is a concentrated theoretical review of scientific research on the stated topic.
The review is done at a good methodological level. positive side of the review.
1. Scientific. Articles, publications listed in PubMed, Embase, PLOS are taken for review. Digital Commons Network, Google Scholar and Cochrane Library
2. Objectivity. The authors selected the article for review according to a clear algorithm. They took articles where the terms 'hormesis', 'hormesis in aging', 'hormesis in health', 'germline survival', 'neuronal stress response', and 'neuron/germline communication' were encountered.
3. Modern digital technologies were used for the analysis. they used a generative pre-trained transformer (ChatGPT-4) to generate a small amount of original background text, in less than 5% of the paper
The authors' article showcases the result of an interesting experiment on digital text analysis performed by a person.
Remarks.
It should be noted that the phenomenon of Hormesis was introduced by scientists studying the positive effect of low doses of radiation. However, this effect was not subsequently confirmed. And now it is generally accepted in science that the impact of any doses of radiation is negative.
The authors need to somehow reflect this side of the analyzed phenomenon in their review. Now they only talk about the good side of Hormesis
It is desirable to emphasize this aspect in all sections of the review. If there are no such publications among the analyzed articles, then note the absence of data on the negative effect of Hormesis.
Author Response
We would like to thank the reviewer for the input and comments. Hormesis has been a controversial subject but, increasingly, there is new research showing its effects beyond the field of radiation or toxicology. Many laboratory examples of hormesis have been described over the past decades, but there are also new studies which describe its clinical properties, and also its effects on the ageing process. We have added 15 new studies in this respect, in order to strengthen the suggestion that hormesis has very relevant consequences in the subjects we discuss.
We also modified some of the text in the hormesis section, to reflect these facts, as suggested by the reviewer.
15 new studies added
- Calabrese, E.J.; Calabrese, V. Enhancing health span: muscle stem cells and hormesis. Biogerontology 2022; 23(2):151-167
- Calabrese, E.J.; Hormesis Mediates Acquired Resilience: Using Plant-Derived Chemicals to Enhance Health. Annu Rev Food Sci Technol 2021; 12:355-381
- Patrick, R.P.; Johnson, T.L. Sauna use as a lifestyle practice to extend healthspan. Exp Gerontol 2021; 154:111509
- Ji, L.L.; Kang, C.; Zhang, Y. Exercise-induced hormesis and skeletal muscle health. Free Radic Biol Med 2016; 98:113-122
- Cook, R.; Calabrese, E.J. The importance of hormesis to public health. Cien Saude Colet 2007; 12(4):955-63
- Mattson, M.P. Dietary factors, hormesis and health. Ageing Res Rev 2008; 7(1):43-8
- Bukowski, J.A.; Lewis, R.J. Hormesis and health: a little of what you fancy may be good for you. South Med J 2000; 93(4):371-4
- Poumadère, M. Hormesis: public health policy, organizational safety and risk communication. Hum Exp Toxicol 2003; 22(1):39-41; discussion 43-9
- Rattan, S.I. Ageing, gerontogenes, and hormesis. Indian J Exp Biol 2000; 38(1):1-5
- Forcina, L.; Franceschi, C.; Musarò, A. The hormetic and hermetic role of IL-6. Ageing Res Rev 2022; 80:101697
- Lajqi, T.; Stojiljkovic, M.; Wetzker, R. Toxin-induced hormesis may restrain aging. Biogerontology. 2019; 20(4):571-581.
- Martel, J.; Chang, S.H.; Wu, C.; Peng, H.H.; Hwang, T.L.; Ko, Y.F.; Young, J.D.; Ojcius, D.M. Recent advances in the field of caloric restriction mimetics and anti-aging molecules. Ageing Res Rev 2021; 66:101240
- Calabrese, E.J.; Iavicoli, I.; Calabrese, V. Hormesis: why it is important to biogerontologists. Biogerontology 2012; 13(3):215-35
- Chirumbolo, S. Possible role of NF-κB in hormesis during ageing. Biogerontology 2012; 13(6):637-46
- Jacome Burbano, M.S.; Gilson, E. The Power of Stress: The Telo-Hormesis Hypothesis. Cells 2021; 10(5):1156
Reviewer 2 Report
The article entitled “The impact of hormesis, neuronal stress response, and reproduction, upon clinical ageing: a narrative review” aims to delve into the intricate association of hormesis between neurons and germ cells within the context of human ageing. The review extensively discusses various facets of clinical ageing, including hormesis, autophagy, environmental enrichment, neuronal stress response, and the interplay between neurons and germline. However, it should be noted that this endeavor is inherently limited in its scope, and certain aspects may remain inadequately addressed.
Despite attempting to consolidate recent literature, the article lacks the essential element of novelty.
The English language utilized in the article is of good quality. The authors exhibit a strong grasp of grammar and syntax, which minimizes any potential confusion or ambiguity in sentence structure.
Author Response
We thank the reviewer for the comments. We agree that some of the subjects discussed, ideally may need more extensive explanations, however we feel this would be beyond the scope of our intentions. Our aim is to describe the mechanisms and provide some background of a new biology which is beyond the conventional (reproduction, leading to ageing and death). This new biology is based on the complexity of cognition leading to indefinite survival without chronic degenerative diseases. It represents a shift from a physical model of ageing to a more cognitive one. We believe that is the essential novelty of our paper, albeit the first step towards our aim.
We have modified some text in order to reflect and clarify this.
There is some discrepancy in the reviewer’s comments about the language:
On the ‘Quality of English Language’ section, the option X is chosen
[ (x) Moderate editing of English language required]
But this is not reflected in the “Comments on the Quality of English Language” section
In any case, we used a professional in order to confirm and/or correct the text.
Round 2
Reviewer 2 Report
I would like to thank the authors for considering my comments. The authors have made significant improvements to the manuscript and answered queries about the novelty of the subject.
The English language utilized in the article is of good quality.